# A Vibrissa-Inspired Highly Flexible Tactile Sensor: Scanning 3D Object Surfaces Providing Tactile Images

**DOI:** 10.3390/s21051572

**Published:** 2021-02-24

**Authors:** Lukas Merker, Joachim Steigenberger, Rafael Marangoni, Carsten Behn

**Affiliations:** 1Technical Mechanics Group, Technische Universität Ilmenau, 98693 Ilmenau, Germany; 2Institute of Mathematics, Technische Universität Ilmenau, 98693 Ilmenau, Germany; joachim.steigenberger@tu-ilmenau.de; 3Institute for Process Measurement and Sensor Technology, Technische Universität Ilmenau, 98693 Ilmenau, Germany; rafael@rrmarangoni.com; 4Faculty of Mechanical Engineering, Schmalkalden University of Applied Sciences, 98574 Schmalkalden, Germany

**Keywords:** vibrissa, bio-inspired sensor, surface scanning, object reconstruction

## Abstract

Just as the sense of touch complements vision in various species, several robots could benefit from advanced tactile sensors, in particular when operating under poor visibility. A prominent tactile sense organ, frequently serving as a natural paragon for developing tactile sensors, is the vibrissae of, e.g., rats. Within this study, we present a vibrissa-inspired sensor concept for 3D object scanning and reconstruction to be exemplarily used in mobile robots. The setup consists of a highly flexible rod attached to a 3D force-torque transducer (measuring device). The scanning process is realized by translationally shifting the base of the rod relative to the object. Consequently, the rod sweeps over the object’s surface, undergoing large bending deflections. Then, the support reactions at the base of the rod are evaluated for contact localization. Presenting a method of theoretically generating these support reactions, we provide an important basis for future parameter studies. During scanning, lateral slip of the rod is not actively prevented, in contrast to literature. In this way, we demonstrate the suitability of the sensor for passively dragging it on a mobile robot. Experimental scanning sweeps using an artificial vibrissa (steel wire) of length 50 mm and a glass sphere as a test object with a diameter of 60 mm verify the theoretical results and serve as a proof of concept.

## 1. Introduction

The increasing prevalence of robotic devices in industry, as well as in everyday life, leads to growing challenges in interactions between robots and their environments. One fundamental requirement for robots is the ability to move safely in the environment, i.e., avoiding unexpected collisions. If a robot is too insensitive due to a lack of appropriate exteroceptive sensors [1], the environment always presents a hazard to the robot, or worse, the robot poses a risk to the environment. Moreover, most robotic applications include some kind of interaction with external objects, requiring the knowledge of various information, such as object positions, orientations, shapes and surface normals. This environmental information is frequently provided by optical sensors. However, a robot relying solely on computer vision might experience a loss of information, when external obstacles or dirt occlude its cameras, or when operating in harsh and a noisy or reflecting environment. Under these conditions, tactile sensor systems would be advantageous to complement or temporarily replace optical systems.

Developing tactile sensors, engineers often draw their inspiration from biology. For instance, a large number of robots and grippers is equipped with tactile sensors inspired by the human skin [2,3]. One drawback of these sensors is their limited scanning range. A mobile robot whose surface is equipped with skin-like sensors would have to be in direct contact with an object in order to sense a point, subsequently moving backwards to avoid the obstacle, see Figure 1a. However, such complicated trajectories are often undesired and increase operation time.

Therefore, many robots would benefit from tactile sensors with a near-field scanning range, as sketched in Figure 1b. In this scenario, the robot might initially move through the environment without any sensor-object contact. After the tactile sensor accidentally makes contact with some point of the environment, it is bent, causing the support reactions to exceed a certain activation threshold. Subsequently, an object path-following algorithm might be used, tracking a prescribed value of the support reactions to achieve an optimal scanning movement of the robot, relative to the object.

Besides the human sense of touch, more inspiration for developing tactile sensors can be found in animals: one prominent and particularly well-researched sense organ is the mystacial vibrissae of a rat, providing information about object distances, orientations, shapes and textures [4,5,6]. Basically, a vibrissa consists of a long, slender, conical and pre-curved hair-shaft [7,8] with no receptors along its length, which is supported by its own follicle-sinus complex (FSC). Making contact with an object, mechanical stimuli are transmitted to the FSC, where actual sensing takes place [9]. Even though it is not conclusively clarified which mechanical quantities, the animals rely on and how they determine contact positions in space [10], the morphological structure of a vibrissa often serves as a paragon for developing tactile sensors. Within this paper, we focus on sensor models for object shape scanning and reconstruction.

### 1.1. Vibrissa-Based Sensors

In the literature, vibrissa-based sensors for object shape scanning and reconstruction often consist of a single artificial vibrissa-shaft, which is modeled as a long and slender flexible rod, clamped at one end (foot, base, support), and free at the other end (tip). In order to sense a point in space, the rod is usually brought into contact with an object of interest by a rotational or translational movement of its support. As the rod makes contact, it is bent and different mechanical signals at its base are measured and evaluated to draw conclusions about the contact position in space. In doing so, two fundamentally different scanning strategies prevailed in the literature [11], see Figure 2:Tapping is a scanning approach of repeatedly rotating or translating the rod against an object by small pushing angles and immediately retracting it from the object (right after the initial contact). Each tap provides only one or a small number of contacts and must, therefore, be repeated to sense various points of the object. Since tapping is characterized by small deformations of the rod, a slightly flexible or approximately rigid rod is sufficient for object scanning.Sweeping is a method of rotating or translating a rod against an object far beyond the initial contact, causing the rod to slide over the object’s surface. In this way, a single sweep leads to a variety of sensed contact points revealing a large part of the object. The sweeping scanning strategy is characterized by large deformations and therefore requires a highly flexible rod.

The first approach is usually realized by rotational scanning movements [12,13,14,15]. Due to small deformations, the problem of contact point localization is addressed using linear bending theory, what greatly simplifies the reconstruction procedure. In these cases, angle and moment information at the base of a rod or even the time derivatives of these quantities [15] are sufficient to determine the contact position in space. Instead of moment information, the deformation angle of the rod is evaluated [13]. In [16], the authors stated that the support reactions of a one-sided clamped rod in contact with an object are sufficient to determine the magnitude and the line of action of the unknown contact force, regardless of using a rigid or a flexible rod. For a rigid rod, the contact position in space uniquely results from the intersection between its axis and the reconstructed line of action. In [17], this approach was adapted for using a flexible rod, estimating its elastic line using linear theory.

One common drawback of all tapping approaches is that they require actuators for both making tapping movements and repeatedly changing the position of the rod relative to the object in order to achieve the complex scanning kinematics, see Figure 2a. Such a complex approach, imitating the head and vibrissae movements of a rat, is described in [18]. However, the sweeping scanning strategy dispenses with the need for various actuators, due to the fact that sweeping can be realized passively. For example, if a mobile robot equipped with a highly flexible artificial vibrissa passes an object, the robot movement itself causes the artificial vibrissa to slide along the object, consequently providing a whole sequence of contact points, see Figure 1b. One positive side effect of using a highly flexible rod is its robustness against collisions and the ability to bend out of the way when accidentally making contact with an obstacle. Despite these advantages, the sweeping strategy has received much less attention in the literature. In [19], a super-elastic nitinol wire was swept along an object using a DC motor. The support reactions at the base of the wire were measured using a hub load cell and used to repeatedly reconstruct the entire shape of the wire, finally making the 2D contour of the scanned object apparent. In [20,21], the problem of sweeping a rod along 2D objects was treated analytically. Beyond object reconstruction, the focus was on simulating scanning sweeps and generating the support reactions theoretically. As in [19], nonlinear Euler–Bernoulli theory was used for describing the large deformations of the rod. An alternative reconstruction method was presented in [11]. The algorithm is based on repeatedly inferring from one contact point to the next one by continuous measurement of the moment and rotation angle at the base of the rod. One major advantage of this method is that no force measurement is required, consequently reducing the size of the sensors. Furthermore, the contact points are determined by solving algebraic equations, instead of numerically integrating the deformation equations of the rod, which greatly reduces computational time. However, the method is limited to tangential contacts along the length of the rod. Tip contacts, which are likely to occur in real-world problems, lead to large reconstruction errors. In addition, it requires an active movement and was never extended for scanning 3D objects.

Considering object scanning as a plane problem is only permitted if certain assumptions are made regarding the shape and orientation of the object with respect to the scanning plane. The latter is defined as a virtual plane in which the rod would freely rotate or translate through space due to the scanning displacement but without any object contact. In order to clarify under which conditions a scanning sweep can be modeled and treated as a plane problem, we consider the example of a cylinder—see Figure 3.

The scanning plane (gray) intersects the cylinder in a curve (red line). For a special arrangement of the scanning plane relative to the object, the surface normals at each point of the intersection curve lie within the scanning plane, see Figure 3b. Under these conditions, the scanning sweep entirely takes place in a plane due to the fact that, in the absence of friction (ideal contact), the contact force between the rod and the object coincides with the normal vector at the contact point. However, it is clear that these special conditions are practically unlikely to occur and can only be ensured if the orientation of the scanned object is a priori known. This may be the case for some test stands or prototypes as, e.g., presented in [22], but not in a real-world scanning scenario. For the latter, it is much more likely that the rod makes contact with the object in some random orientation and, therefore, bends out of the scanning plane. In addition, real-world objects might be shaped in such a complex way that, even, if the orientation of the object was known, the special case, shown in Figure 3b, cannot be achieved without actively adjusting the scanning direction. To sum up, using 2D models for object scanning greatly limits the practical applicability.

Considering the process of a rod making contact with an object as a spatial problem, the authors in [11] state that there are two distinct ways that a whisker can slip along a surface: lateral and longitudinal slip. Longitudinal slip occurs when the rod slides over the object’s surface within the scanning plane, while lateral slip occurs when the rod slides out of the scanning plane. Simultaneously, the contact point might change along the rod itself, which is referred to as axial slip. However, the authors in [11] discuss the possibility of the occurrence but limit their analysis to a plane scenario without lateral slip, as, e.g., shown in Figure 3b. Another approach considering 3D object scanning including the large deformations of the rod is presented in [23]. There, an arbitrary orientation of the scanned object is permitted, but the lateral slip of the rod is prevented by actively adjusting the scanning direction (and thus the scanning plane), in a way that the rod is always pushed against the normal direction of the object. However, this procedure is rather impractical for robotic applications as it can neither be used passively, e.g., dragging the sensor on a mobile robot nor can it be implemented using multiple artificial vibrissae with a single actuator.

### 1.2. Objective of the Present Work

The paper at hand contributes to the overall goal of scanning and reconstructing 3D object shapes by means of vibrissa-inspired sensors, providing tactile images. Here, the object scanning process is defined as an experimental or simulated scanning sweep of an artificial vibrissa along a prescribed object with the aim to measure or theoretically generate the mechanical signals at the base of the vibrissa. In contrast, object reconstruction denotes the inverse process of using the simulated or measured mechanical signals at the base of the artificial vibrissa in order to draw conclusions about the contact position in space. Considering both steps, we aim to demonstrate the general functionality of the presented tactile sensor principle using a highly flexible rod as a transducer/probe. Focusing on a proof of concept, the study does not yet include aspects such as path planning algorithms, nor an optimal placement of the presented sensor on a robot’s surface. Although we choose a vibrissa as a paragon, we do not aim to copy a single vibrissa in as detailed as possible a manner, or to gain new insights into the contact localization process of the rat. Instead, we propose a technical sensor principle, in which some of the sensor properties are inspired by vibrissae.

In Section 2, we particularly address the following sub-goals: In Section 2.1 we present a model consisting of a single, one-sided clamped, artificial vibrissa, which is modeled as a straight, cylindrical Euler–Bernoulli bending rod. In order to achieve scanning sweeps along an object of interest, the clamping position of the rod is shifted quasi-statically along a (straight) scanning trail. As a consequence, the rod slides along the object, undergoing large bending deflections and causing measurable support reactions at the clamping. In contrast to previous works, our focus is not solely on using these support reactions to reconstruct the object’s shape. Instead, we include the inverse process of simulating scanning sweeps along a known object surface, theoretically generating the support reactions. In this way, the study gains new insights into how a rod slides along a 3D surface, permitting longitudinal and lateral slip and distinguishing between tip and tangential contact. Finally, we demonstrate how the support reactions at the base of the rod can be used to reconstruct surface points and corresponding surface normals. In Section 2.2, we present an experimental setup, which is used to validate the mechanical model, consisting of a test object and a spring steel wire attached to a force-torque transducer. In the first part of Section 3, the simulated support reactions are compared with measured data. In the second part, we determine clouds of contact points based on the measured and simulated support reactions and use these data to reconstruct 3D surfaces.

## 2. Materials and Methods

### 2.1. Modeling

#### 2.1.1. Modeling the Isolated Artificial Vibrissa

The artificial vibrissa is modeled as a circular cylindrical rod, consisting of homogeneous and isotropic Hooke’s material. Thus, its shape is characterized by the length *L* and the constant circular cross-section (with constant second moment of area *I*). Its mechanical behavior is essentially determined by a constant Young’s modulus *E*, resulting in a constant bending stiffness EI.

**Remark** **1.**
*From the outset, we make the following agreement regarding the units of measure based on [20]*
(1)$$\begin{array}{c} \left[length\right]:=L\;\left[force\right]:=\frac{EI}{L^{2}}\;\left[moment\right]:=\frac{EI}{L} \end{array}$$
*Any relation in the realm of artificial vibrissae sketched above now formally contains only dimensionless system parameters. Therefore, the findings of the present paper (focusing on artificial sensors) might be transferred to real world vibrissae, supposing that all technical assumptions are fulfilled.*


Supposing one end of the rod (“foot, base”) clamped and some other single point of the rod in contact with an object, the rod gets bent in the inference of some still arbitrary contact force, forming an elastic line in $R^{3}$. However, due to the above working hypothesis of homogeneity and isotropy, the elastic line shrinks to one in a plane $E_{\psi }$, which is fixed by the clamp and the contact force. This fact is exploited to keep the model equations simple and the computing time of the used algorithm as low as possible. Let, in a fixed (x,y,z)-coordinate system of $R^{3}$, the clamp be given at $P_{0}=\left(x_{0},y_{0},0\right)$ with constraint direction $\varphi_{0}=\frac{\pi }{2}$. Assume, for now, the plane $E_{\psi }$ is known. We then introduce body-fixed Cartesian coordinates (u,v,w) with $P_{0}$ at the origin and come up with the scenario sketched in Figure 4.

Thus, we have the transformation rules:(2)uvw=cos(ψ)−sin(ψ)0001sin(ψ)−cos(ψ)0x−x0y−y0z=:T(ψ)·x−x0y−y0z
and
(3)xyz=x0y00+cos(ψ)0−sin(ψ)sin(ψ)0−cos(ψ)010·uvw=x0y00+T−1(ψ)·uvw.

In addition, let us define the angle $\alpha \in (-\frac{\pi }{2},\frac{\pi }{2})$ as the signed angle by which $-{\vec{e}}_{v}$ must rotate around ${\vec{e}}_{w}$ to reach $\vec{f}$—see Figure 4. Therefore, α is the orientation angle of the contact force $\vec{f}$ within the deformation plane $E_{\psi }$:(4)$$\begin{array}{c} \alpha =\mathrm{atan}2({\vec{e}}_{u}\cdot \vec{f},-{\vec{e}}_{v}\cdot \vec{f}), \end{array}$$
where atan2 denotes the four-quadrant inverse tangent, as frequently used in numerical software, see MATLAB documentation: https://de.mathworks.com/help/matlab/ref/atan2.html (accessed on 14 December 2020).

In $E_{\psi }$, the elastic line writes (u(s),v(s),φ(s)), where *s* is the arc length and φ is the slope angle of the rod. Following the normalization (Equation 1) (see Remark 1), we have to treat the Euler–Bernoulli bending equation
(5)$$\begin{array}{ccccc} \frac{du(s)}{ds}=\cos\left(\varphi \left(s\right)\right) & \; & \frac{dv(s)}{ds}=\sin\left(\varphi \left(s\right)\right) & \; & \frac{d\varphi (s)}{ds}=\kappa \left(s\right)=m\left(s\right) \end{array}$$
with s∈(0,1) and curvature κ equal to bending moment (mind, e.g., ${\vec{f}}_{0}=\frac{EI}{L^{2}}\cdot \vec{f}$, with dimensionless $\vec{f}=f\;\left(\sin\left(\alpha \right)\;{\vec{e}}_{u}-\cos\left(\alpha \right)\;{\vec{e}}_{v}\right)$). The corresponding boundary conditions (BCs) are to be adjoined—see Section 2.1.3.

#### 2.1.2. Modeling the Scanned Surface

Real-world objects might have various, complex shapes and textures and may be of any compliance. Within the present paper, we model the scanned object as a rigid body with an oriented smooth and regular surface z=C(x,y) [24]. Hereafter, we always consider that side of the surface facing the *x*-*y*-plane. In doing analysis, we use the following notation for brevity: Given a continuous function (x,y)↦f(x,y), then, $f_{,x}\left(x,y\right):=\frac{\partial }{\partial x}f\left(x,y\right)$ denotes the partial derivative of *f* with respect to *x*.

Each point *P* of the surface *C* is defined by the position vector:(6)$$\vec{r}\left(x,y\right)=x{\vec{e}}_{x}+y{\vec{e}}_{y}+C\left(x,y\right){\vec{e}}_{z}$$

Using the local frame
(7)$$\begin{array}{c} \begin{array}{c} \vec{r}{,}_{x}\left(x,y\right)={\vec{e}}_{x}+C{,}_{x}\left(x,y\right){\vec{e}}_{z}\;\vec{r}{,}_{y}\left(x,y\right)={\vec{e}}_{y}+C{,}_{y}\left(x,y\right){\vec{e}}_{z} \end{array} \end{array}$$
we find the unit normal vector field of C: (8)$$\begin{array}{c} \vec{n}\left(x,y\right)=\frac{\vec{r}{,}_{x}\left(x,y\right)\times \vec{r}{,}_{y}\left(x,y\right)}{|\vec{r}{,}_{x}\left(x,y\right)\times \vec{r}{,}_{y}\left(x,y\right)|}\\ \;=\frac{1}{\sqrt{{C{,}_{x}}^{2}\left(x,y\right)+{C{,}_{y}}^{2}\left(x,y\right)+1}}\left(\begin{array}{c} -C{,}_{x}\left(x,y\right)\\ -C{,}_{y}\left(x,y\right)\\ 1 \end{array}\right)=:\frac{1}{\sqrt{r(x,y)}}\left(\begin{array}{c} -C{,}_{x}\left(x,y\right)\\ -C{,}_{y}\left(x,y\right)\\ 1 \end{array}\right) \end{array}$$

Even though, some objects of our daily use are bounded by quite complicated overall surfaces, the latter can often be decomposed into smaller simple surface elements. Therefore, scanning an object locally in small scale, it is justifiable to classify surfaces under investigation by considering how they are locally curved, see Appendix A. One problem, which is related to the overall shape of the scanned object, was highlighted in [11] and recently analyzed in more detail in [25]: scanning concave objects might result in simultaneous contacts at multiple distinct points along the rod.

In the present paper, a multi-point contact scenario would violate the model assumption of a single contact force (see Section 2.1.1). In addition, it is the nature of the sensing principle, that concave areas of an object might be over-swept. Given this backdrop, the mechanical model presented in the paper at hand is limited to three types of surfaces with no concave parts, based on the surface classification in Appendix A (see green frame in Figure A2):I:Planar surfaces;II:Parabolic convex surfaces;III:Elliptic convex surfaces.

Due to these limitations, the demand on the tactile sensor is not to capture the entire shape of an object in as much detail as possible, but rather to obtain a rough estimate of it, e.g., complementing optical sensors.

#### 2.1.3. Modeling the Scanning and Reconstruction Process Theoretically

Within this subsection we combine the models of the isolated artificial vibrissa (rod) from Section 2.1.1 with the object surface, modeled in Section 2.1.2. In order to realize a scanning sweep of the rod along a given object surface *C*, the clamping position $P_{0}\left(x_{0},y_{0},0\right)$ of the rod is shifted translationally on a given straight trail in the *x*-*y*-plane, see Figure 5.

Assuming this process to be slow enough, the problem is treated as a quasi-static one. After the very first contact, the rod slides along the object undergoing large bending deflections in 3D space. Frictional effects between the rod and the object are neglected ($\vec{f}\|{\vec{n}}_{1}$) for the theoretical model but are discussed later on in Section 3. During scanning, the deformation of the rod results in a set of six support reactions, namely the forces $f_{0x}$, $f_{0y}$, $f_{0y}$ and moments $m_{0x}$, $m_{0y}$, $m_{0z}$. Firstly, we analyze the problem of determining the elastic line and the support reactions of the rod in contact with a prescribed surface for a given clamping position $P_{0}$. Afterwards, we demonstrate that the support position and reactions, which might be known from simulations or measurements, are sufficient to determine both the contact position in 3D space as well as the surface normal at the contact point.

##### Step 1—Determining the Elastic Line and the Support Reactions of the Rod

In a first step, we consider the object surface as given. In the absence of friction, the contact force $\vec{f}$ during scanning is orientated in direction of the outward-pointing normal unit vector ${\vec{n}}_{1}$ at the contact point $P_{1}\left(\xi ,\eta ,\theta \right)$ on *C*, see Figure 5 (the case in the presence of friction is discussed later on in Section 3). Due to the assumptions in Section 2.1.1, the rod bends in some unknown deformation plane $E_{\psi }$ (geometric location of the elastic line) containing the clamping position $P_{0}$, the contact position $P_{1}$ as well as the clamping and force directions ${\vec{e}}_{z}$ and ${\vec{n}}_{1}$, respectively:(9)$$\left\{P_{0},P_{1},{\vec{e}}_{z},{\vec{n}}_{1}\right\}\in E_{\psi }$$

Evaluating the conditions $\left\{P_{0},{\vec{e}}_{z}\right\}\in E_{\psi }$, we find the equation of the plane
(10)$$-\sin\left(\psi \right)\left(x-x_{0}\right)+\cos\left(\psi \right)\left(y-y_{0}\right)=0$$
and the normal vector of $E_{\psi }$
(11)$${\vec{e}}_{\psi }=-\sin\left(\psi \right){\vec{e}}_{x}+\cos\left(\psi \right){\vec{e}}_{y}$$
depending on an unknown orientation angle ψ∈(−π,π], see Figure 5. This plane intersects the surface *C* on some curves $C_{\psi }=E_{\psi }\cap C$ (dashed line in Figure 5). With a view on Equation (Equation 9) the contact point $P_{1}$ is necessarily localized on the intersection curve $C_{\psi }$: (12)$$P_{1}\left(\xi ,\eta ,\theta \right)\in C_{\psi }$$(13)$$\begin{array}{cc} ⟹\; & \eta =y_{0}+\left(\xi -x_{0}\right)\tan\left(\psi \right) \end{array}$$(14)$$\begin{array}{cc} ⟹\; & \psi \left(\xi ,\eta \right)=\mathrm{atan}2\left(\eta -y_{0},\xi -x_{0}\right) \end{array}$$

In addition, according to Equation (Equation 9), $E_{\psi }$ is a normal plane at $P_{1}$:$${\vec{n}}_{1}\in E_{\psi }\Leftrightarrow {\vec{n}}_{1}⊥{\vec{e}}_{\psi }\Leftrightarrow {\vec{n}}_{1}\cdot {\vec{e}}_{\psi }=0$$

Using Equations (Equation 8) and (Equation 11), we find the condition:(15)$$C{,}_{x}\left(\xi ,\eta \right)\sin\left(\psi \right)-C{,}_{y}\left(\xi ,\eta \right)\cos\left(\psi \right)=0$$
with the unknown parameters ξ, η and ψ. Substituting ψ using Equation (), we end up in an implicit function (where $\left(x_{0},y_{0}\right)$ enters as parameters) of the form
(16)$$F\left(\xi ,\eta \right):=C{,}_{x}\left(\xi ,\eta \right)\left(\eta -y_{0}\right)-C{,}_{y}\left(\xi ,\eta \right)\left(\xi -x_{0}\right)=0$$

For a given clamping position $P_{0}\left(x_{0},y_{0},0\right)$, only those points of the prescribed surface *C*, that fulfill Equation (Equation 16), are optional contact points. Using Euler’s constitutive law, the plane deformation of the rod is described by the following system of ordinary differential equations (ODE), where *s* is the natural coordinate arc length, φ is the slope of the rod’s axis and κ is the curvature of the rod [20]:(17)$$\begin{array}{c} \begin{array}{c} \begin{array}{cccc} \frac{du(s)}{ds} & =\cos(\varphi (s))\; & \frac{d\varphi (s)}{ds} & =\kappa (s)\\ \frac{dv(s)}{ds} & =\sin(\varphi (s))\; & \frac{d\kappa (s)}{ds} & =f\cos(\varphi (s)-\alpha ) \end{array} \end{array} \end{array}$$

Based on [20,22], two contact phases have to be distinguished: Tip contacts with a known contact position $s_{1}=1$ but an unknown contact angle φ(1)>α, and tangential contacts with an unknown contact position $s_{1}\in \left(0,1\right]$ but a known contact angle: $\varphi (s_{1})=\alpha$. Using Equations (Equation 8), (Equation 14), (Equation 2) and (Equation 4), we find the following BCs: (18)$$\begin{array}{cc} \mathrm{tip}\;\mathrm{contacts}:\; & \begin{array}{c} \begin{array}{cccc} \; & u(0)=0\; & u(1) & =u_{1}\left(\xi ,\eta \right)\\ \; & v(0)=0\; & v(1) & =v_{1}\left(\xi ,\eta \right)\\ \; & \varphi \left(0\right)=\frac{\pi }{2}\; & \kappa (1) & =0 \end{array} \end{array} \end{array}$$(19)$$\begin{array}{cc} \mathrm{tangential}\;\mathrm{contacts}:\; & \begin{array}{c} \begin{array}{cccc} \; & u(0)=0\; & u(s_{1}) & =u_{1}\left(\xi ,\eta \right)\\ \; & v(0)=0\; & v(s_{1}) & =v_{1}\left(\xi ,\eta \right)\\ \; & \varphi \left(0\right)=\frac{\pi }{2}\; & \varphi (s_{1}) & =\alpha (\xi ,\eta )\\ \; & \; & \kappa (s_{1}) & =0 \end{array} \end{array} \end{array}$$

Both boundary-value problems Equations (Equation 17) and (Equation 18), and Equations (Equation 17) and (Equation 19) contain the unknown parameters ξ, η, *f*. In addition, Equations (Equation 17) and (Equation 19) contain the unknown contact position $s_{1}$ but also one additional BC for the contact angle. Using the implicit function (Equation 16) as an additional purely geometrical condition, both BVPs are well defined.

**Remark** **2.***Depending on C, the implicit function Equation* (Equation 16) *might not be globally uniquely solvable [26], but can be exploited numerically. However, later on, we consider some special cases of C, which allow to us rewrite Equation* (Equation 16) *as an explicit function g:ξ↦η=g(ξ). Thus, the contact coordinate ξ defines the contact point $P_{1}\left(\xi ,\eta \left(\xi \right),C\left(\xi ,\eta \left(\xi \right)\right)\right)$ as well as the corresponding normal vector ${\vec{n}}_{1}$ using Equation* (Equation 8).


Once the parameters ξ, η, *f* and $s_{1}$ are known, the support reactions with respect to the plane $E_{\psi }$ are determined in the following way
(20)$$\begin{array}{c} f_{0u}=-f\cdot \sin\left(\alpha \right),\;f_{0v}=f\cdot \cos\left(\alpha \right),\;m_{0w}=u_{1}\cos\left(\alpha \right)+v_{1}\sin\left(\alpha \right) \end{array}$$
and, finally, transformed to the global coordinate system using Equation (Equation 3):(21)$$\begin{array}{c} \left(\begin{array}{c} f_{0x}\\ f_{0y}\\ f_{0z} \end{array}\right)=T^{-1}\left(\psi \right)\cdot \left(\begin{array}{c} f_{0u}\\ f_{0v}\\ 0 \end{array}\right),\;\left(\begin{array}{c} m_{0x}\\ m_{0y}\\ m_{0z} \end{array}\right)=T^{-1}\left(\psi \right)\cdot \left(\begin{array}{c} 0\\ 0\\ m_{0w} \end{array}\right) \end{array}$$

Note that the twist moment $m_{0z}$ is always zero due to the plane-bending assumption.

#####  Step 2—Contact Point Localization Based on the Support Reactions

For the actual sensor application, we assume the support reactions (i.e., the observables) at the base of the rod as well as the support position to be known by measurements or simulations:(22)$$\begin{array}{c} \mathrm{known}\;\mathrm{quantities}:\;\begin{array}{c} P_{0}=\left(x_{0},y_{0},0\right),\;{\vec{f}}_{0}=\left(\begin{array}{c} f_{0x}\\ f_{0y}\\ f_{0z} \end{array}\right)\;{\vec{m}}_{0}=\left(\begin{array}{c} m_{0x}\\ m_{0y}\\ m_{0z} \end{array}\right) \end{array} \end{array}$$

In contrast to the inverse problem of generating the support reactions, the orientation angle ψ, defining the deformation plane $E_{\psi }$ directly results from evaluating the support reactions $f_{0x}$ and $f_{0y}$ or $m_{0x}$ and $m_{0y}$, alternatively:(23)$$\psi =\mathrm{atan}2\left(-f_{0y},-f_{0x}\right)=\mathrm{atan}2\left(m_{0x},-m_{0y}\right)$$

In Section 3, it will be shown that the latter alternative of Equation (Equation 23), which is based on the clamping moment, may be beneficial. Using Equations (Equation 2) and (Equation 23) the orientation of the local coordinate system (u,v,w) is determined. In addition, we calculate the magnitude *f* and the direction of the contact force, which coincides with the surface normal ${\vec{n}}_{1}$ in the absence of friction:(24)$$\begin{array}{c} f=|\vec{f_{0}}|=\sqrt{{f_{0x}}^{2}+{f_{0y}}^{2}+{f_{0z}}^{2}}\;\mathrm{and}\;{\vec{n}}_{1}=-\frac{1}{f}\cdot {\vec{f}}_{0} \end{array}$$

Determining α by means of Equation (Equation 4), we have all parameters defining Equation (Equation 17) at hand. Using the known parameters Equation (Equation 22), we formulate an initial-value problem consisting of the ODE-system Equation (Equation 17) and the following initial conditions:(25)$$\begin{array}{c} \begin{array}{c} u\left(0\right)=0\;v\left(0\right)=0\;\varphi \left(0\right)=\frac{\pi }{2}\;\kappa \left(0\right)=-m_{0w} \end{array} \end{array}$$
with $m_{0w}=\sqrt{m_{0x}^{2}+m_{0y}^{2}}$. Then, the contact point $\left(u_{1},v_{1}\right)$ in $E_{\psi }$ results from numerically integrating Equations (Equation 17) and (Equation 25) with the termination condition, that the curvature $\kappa (s_{1})$ at the contact point must be zero, due to the single force assumption, see Section 2.1.1. Finally, the representation of the contact point in $R^{3}$ follows from Equation (Equation 3).

##### Step 3—Object Scanning and Reconstruction by Reiteration of Step 1 and Step 2

Within the present paper, object scanning and reconstruction are two separated subsequent processes implemented in *MATLAB R2019a*. Firstly, object scanning is either simulated or performed in a real experiment in order to capture the support reactions. In doing so, the object reconstruction does not yet happen in real-time. Instead, it is performed after the scanning process (simulation or experiment) is completed and the entire sequences of all support reactions as a basis for object reconstruction are stored.

For simulating object scanning, Step 1 is performed repeatedly for each clamping position $P_{0,i}$ on the given scanning trail (step-size 0.01). In order to detect the contact phase, tangential contact is presupposed (use of Equation (Equation 19)) and the solution is examined for a contradiction in a way that the contact position $s_{1}$ must not exceed the length 1 of the rod: Thus, for $s_{1}\leq 1$, the solution is correct and the rod is in tangential contact with the object. For the special case $s_{1}=1$, the rod is in tangential contact at the tip. If $s_{1}>1$ (contradiction), the rod is in tip contact and the calculation step is repeated with the corresponding BCs. The simulation algorithm aborts if no further equilibrium is found. The results include the contact parameters, as well as the support reactions and the elastic line of the rod during the entire scanning sweep.

For object reconstruction, Step 2 is repeatedly executed at each clamping position along the scanning trail. Thus, the support reactions from one scanning sweep lead to a sequence of reconstructed contact points on the object’s surface. In this way, the object reconstruction based on multiple scanning sweeps leads to a 3D cloud of points accompanied by normal vectors resulting from Equation (Equation 24). This provided database is very similar to the one provided by some optical systems, e.g., a laser range finder. Therefore, problems such as fitting a surface to a 3D point cloud accompanied by normal vectors are one of the major challenges in 3D computer vision. In order to transfer the database provided by the tactile sensor to a tactile image, we take advantage of an approach using Implicit B-splines [27] originally meant to assist 3D computer vision.

### 2.2. Experiments

A straight spring steel wire according to DIN EN 10270-1:2017-09 with diameter d=0.3 mm, length L=50 mm and a Young’s modulus E=206 GPa was used as an artificial vibrissa, which was one sided clamped to a multicomponent force and torque transducer [28] (see Figure 6a). The following units of measure result from the chosen material and geometry of the wire:(26)$$\begin{array}{c} {}[length]:=50\;\mathrm{mm}\;[force]:=0.0328\;N\;[moment]:=1.638\;N\;\mathrm{mm} \end{array}$$ It is directly evident from Equation (Equation 1), that the rod parameters *E*, *I* and *L* scale the forces and moments at varying degrees. Therefore, the geometric dimensions of the wire were chosen based on preliminary experiments in order to adapt the mechanical signals at the base of the wire to the measuring range of the force and torque transducer, exploiting the well-known mechanical properties of the chosen steel wire.

The used multicomponent force and torque transducer is a custom-built prototype based on the principle of electromagnetic force compensation [28], see Appendix B for further details about the transducer.

The entire assembly, consisting of the transducer and the attached steel wire, was mounted on a two-axis linear stage, which was movable in the *x*- and *z*-directions (see Figure 6a). A glass sphere with diameter D=60 mm was used as a test object for an in-depth study of object shape scanning and reconstruction.

**Remark** **3.**
*According to Figure A2 (Appendix A), the spherical object was chosen as a striking example of the elliptic convex surface category. In general, other surfaces of this type, e.g., an ellipsoid or paraboloid, might have been chosen as the analytical approach in Section 2.1 which are generally sufficiently. Therefore, a sphere is rather chosen for reasons of commercial availability as well as metrological verifiability than due to any measuring limitations of the presented sensor principle.*


The object was clamped to a support specimen, which was attached to a linear stage in the *y*-direction. In a mobile robotic application, the sensor assembly would be carried/dragged by a robot and, therefore, moved relative to an object of interest in the environment. Due to the ease of implementation, we proceeded the other way around by moving the object and keeping the sensor assembly fixed in space. However, both scanning procedures described are functionally equivalent. Therefore, describing the experimental procedure, we assume the clamping position to move in an object-fixed coordinate system (which better reflects the practical application), keeping in mind that the actual scanning kinematics were inverted.

Each experimental scanning sweep occurred as follows: Initially, the wire was moved out of contact to some start position with a specified distance *h* for object scanning (see Figure 6a,b). The object distance is defined as the smallest distance between the object and the *x*-*y*-plane. Subsequently, the base position $P_{0}\left(x_{0},y_{0},0\right)$ of the rod was moved on a straight scanning trail with some constant $x_{0}$ in the positive *y*-direction (see Figure 6b). After the very first contact between the tip of the undeformed wire and the object, the wire swept along the object’s surface until it finally detached. In this way, multiple scanning sweeps were performed using a constant scanning speed of v=1 mm/s as a compromise between a moderate experiment time and keeping dynamical effects as small as possible.

**Remark** **4.**
*In theory, the scanning sweep is realized by incremental displacements of the clamping, resulting in a sequence of consecutive equilibrium states. Therefore, the process is considered as a quasi-static one. In practice, the scanning sweep is performed using continuous movements of the clamping and not by incremental displacements of the clamping waiting for the stationary state at each position as in [22]. Although the latter procedure would be more consistent with the model assumptions, it is more complicated and time-consuming for a mobile robot from a practical point of view.*


The object distance *h* was limited by a minimum value of 25 mm to avoid collisions of the wire with the specimen support and to avoid the plastic deformation of the wire. A maximum distance of 45 mm was chosen to ensure a sufficient deformation of the wire. Varying the parameters *h* (25 to 45 mm, step-size: 5 mm) and $x_{0}$ (−13 to +13 mm, step-size: 1 mm), resulted in a total of 135 performed scanning sweeps, each lasting 74 seconds. Depending on the object distance *h* and the scanning trail, the sweeps either included tip and tangential contacts or only tip contacts. For lower object distances, most sweeps started with a phase of tip contacts, which at some point turned to a phase of tangential contacts and finally returned to a tip contact phase before the wire detached from the sphere. Lateral slip occurred during all experimental scanning sweeps. Longitudinal slip only occurred for some special cases, namely at the beginning of those scanning sweeps with $x_{0}=0$. In addition, for all scanning sweeps, axial slip occurred when the rod was in tangential contact with the sphere. During object scanning, the set of six support reactions at the base of the wire were measured in the (x,y,z)-coordinate system (see Figure 5) with a sampling rate of 40 Hz. The measured data were filtered by a moving mean of 10th order. Since the clamping position $P_{0}$ of the wire was shifted from the center *S* of the transducer by the distance q= 73.5 mm (see Figure 6a), the measured signals were corrected in the following way:(27)$$\begin{array}{c} {\vec{f}}_{0}={\vec{f}}_{s},\;{\vec{m}}_{0}={\vec{m}}_{s}-q\left({\vec{e}}_{z}\times {\vec{f}}_{s}\right) \end{array}$$
where ${\vec{f}}_{s}$ is the measured force and ${\vec{m}}_{s}$ is the measured moment with respect to the center *S* of the transducer. The reconstruction results were characterized by many outliers, especially at the beginning and at the end of the scanning sweep, where the deformation of the wire was small. This is due to the fact that small deflections lead to small measured support reactions and, thus, a poor signal to noise ratio. The outliers were expunged from the reconstruction result by discarding all those reconstructed points with a clamping moment lower than a certain threshold (approximately 0.15 N mm).

## 3. Results and Discussion

Within this section, the theoretically generated support reactions of scanning sweeps on a variety of scanning trails are compared to the measured data, provided by the setup introduced in Section 2.2. In this way, the mechanical sensor model and the presented procedure of object scanning are validated. Subsequently, both the simulated and measured support reactions are used for object reconstruction, finally creating a tactile image of the scanned object.

### 3.1. Measured Signals during Object Scanning

From a theoretical point of view, there are two distinct ways in which the rod can slide along the sphere, depending on the scanning trail The scanning trail defines the scanning plane, which is the plane defined by the base point $P_{0}$ as well as the trail and clamping directions. If $x_{0}=0$, the trail runs exactly below the center of the sphere. For this particular case, the normal vectors of the sphere at each point on the intersection curve between the scanning plane and the sphere lie within the scanning plane, compare Figure 3b. Thus, scanning the sphere on a trail with $x_{0}=0$ is a special case, where the entire scanning sweep takes place in a plane and, thus, no lateral but only longitudinal slip occurs [22]. Such a scanning sweep, ending with a terminating snap-off of the wire from the sphere is shown in Figure 7a. However, another characteristic way that the rod sweeps along the sphere, which is more likely to occur results when the scanning trail is laterally shifted from the coordinate origin $x_{0}\neq 0$. Then, the normal vectors of the sphere at each point on the intersection curve between the scanning plane and the sphere do *not* lie within the scanning plane—compare to Figure 3a. Such a sweep includes lateral slip in a way that the rod symmetrically bends around the sphere and finally smoothly detaches from the object without any snap-off, see Figure 7b. In Figure 7 tip contacts are colored in red, tangential contacts in green and the straight end of the rod ($s\in [s_{1},1]$) in black. Both sweeps start with a sequence of tip contacts, continued with a tangential contact phase and finally end with another sequence of tip contacts. For scanning sweeps with a larger object distance, only tip contacts occur (not shown).

Figure 8 is a schematic representation of five exemplary scanning trails (dashed lines) A–E. It is intended to assist in interpreting the results in Figure 9. The orange dots indicate the position of the first contact between the tip of the undeformed rod and the sphere. Sweep C ($x_{0}=0$ mm) corresponds to Figure 7a and sweep E ($x_{0}=+13$ mm) to Figure 7b.

A comparison between the simulated and experimental support reactions (observables) during object scanning with an object distance of h=25 mm is shown in Figure 9 ((x,y,z)-coordinate system). The rows A–E correspond to the scanning trails A–E in Figure 8. Appearing as smooth curves, the simulated data in Figure 9 can be clearly distinguished from the noisy measured signals. Figure 9 contains the reaction forces $f_{0x}$ (green), $f_{0y}$ (orange) and $f_{0z}$ (blue) on the left as well as the reaction moments $m_{0x}$ (green), $m_{0y}$ (orange) and $m_{0z}$ (blue) on the right. Let us firstly discuss these results from a theoretical point of view and, thus, focus on the simulated data (smooth curves). For all scanning sweeps, except the special case C (see Figure 7a), the support reactions are vertical or point symmetric, as a consequence of the symmetric bending of the rod around the sphere, see Figure 7b. It is obvious that the maximum values of all support reactions increase with decreasing $|x_{0}|$, i.e., the closer the scanning trail is to the coordinate origin. Negative values $x_{0}<0$ (trail A and B) lead to positive reaction forces $f_{0x}$ while positive values $x_{0}>0$ (trail D and E) cause negative forces $f_{0x}$. The component $f_{0y}$ always changes in sign from plus to minus, due to the scanning direction. The reaction force $f_{0z}$ must accept positive values to counteract the contact force. Due to the symmetry of those scanning trails with the same $|x_{0}|$, i.e., A and E as well as B and D (see Figure 8), the components $f_{0y}$ and $f_{0z}$ of the corresponding sweeps are identical while $f_{0x}$ is mirrored with respect to the $y_{0}$-axis. The components $m_{0x}$ and $m_{0y}$ of the clamping moment are positive for negative values $x_{0}<0$ (trail A and B) and vice versa (trail D and E). Again, this leads to a symmetry in the reaction moments of the sweeps A and E and B and D, respectively. According to Equation (Equation 21) the reaction moment $m_{0z}$ must always be zero in the absence of friction (in theory).

Considering the measured data, there is a good qualitative overall accordance with the simulated data. While the maximum values of the support reactions $f_{0x}$, $f_{0z}$, $m_{0y}$ and $m_{0z}$ match well with the simulated data, the maximum values of the components $f_{0y}$ and $m_{0x}$ differ slightly from the simulation. In addition, it is striking that, compared to the simulated data, all measured support reactions, except the reaction force $f_{0z}$, are distorted in the $y_{0}$-direction. Both the deviation of the maximum values and the distortion of the signals are probably caused by frictional effects during the experiment. Another observation, which is likely to be friction-related, is the deviation of the measured component $m_{0z}$ from the theoretically expected value of zero. This is due to the fact that the wire has some radial dimension and, thus, the frictional force induced a twisting moment during lateral slip. In addition, the experiments were influenced by stick-slip effects, which lead to saw-tooth patterns in the support reactions, in particular during scanning sweep D. It was empirically observed that stick-slip effects particularly occurred during lateral slip. Moreover, stick-slip effects were reinforced the closer the trail was to the coordinate origin, $|x_{0}|\to 0$. For longitudinal slip during scanning sweep C, however, stick-slip effects were barely noticeable with the naked eye. It is interesting that the support reactions of the symmetric scanning sweeps B and D differ in a way that scanning sweep D seems to be more affected by stick-slip effects. This indicates that minor external factors, such as soiling or the microscopic surface structure of the wire or the glass sphere strongly affect the stick-slip behavior.

One scanning sweep, particularly standing out of Figure 9 and requiring closer consideration, is the special scanning sweep C. At the beginning of this sweep, the measured support reactions coincide well with the simulated data. However, at some point $y_{0}\approx 15$ mm, all support reactions jump from a high to a lower level. This discontinuity of the experimental data was caused by a lateral jump of the rod from one side to the opposite side of the sphere. In order to understand this behavior, Equations (Equation 14) and (Equation 16) are adapted for the special case of a spherical object:(28)$$\begin{array}{c} \xi =\frac{x_{0}}{y_{0}}\eta \\ \psi =\mathrm{atan}2\left(\eta -y_{0},\frac{x_{0}}{y_{0}}\left(\eta -y_{0}\right)\right) \end{array}$$

Both arguments of Equation (Equation 28) have the same sign, if $\frac{x_{0}}{y_{0}}>0$ and differ in sign if $\frac{x_{0}}{y_{0}}<0$. Depending on η, the term $\eta -y_{0}$ can accept positive or negative values. Therefore, according to the definition of *atan2*, there are two solutions $\psi_{1}$ and $\psi_{2}=\psi_{1}+\pi$ for each clamping position except for $x_{0}=y_{0}=0$, where the second argument of Equation (Equation 28) is not defined. This indicates that, except for $x_{0}=y_{0}=0$, there might be two possible equilibrium states of the rod. For the entire scanning sweep C $\left(x_{0}=0\right)$ the second argument of Equation (Equation 28) is zero, resulting in two possible orientation angles $\psi_{1}=-\frac{\pi }{2}$ and $\psi_{2}=+\frac{\pi }{2}$ of $E_{\psi }$, according to the definition of *atan2*. Due to the principle of minimum energy, it can be expected that the lower potential energy state has a preferential appearance in reality. However, the simulation algorithm proceeds by using the parameters of the previous deformation state as initial values for determining the subsequent deformation state. Therefore, the numerical algorithm always finds that solution, which is the most similar to the previous one and not necessarily that one that minimizes the potential energy. For that reason, the orientation $\psi_{1}=-\frac{\pi }{2}$ does not change during simulation. It can be expected, that, in an ideal world without any disturbance, sweep would occur without a jump, just as observed in the simulation. In practice, however, $x_{0}$ has never the exact value of zero due to minimal deviations. These deviations, however minor, might cause the rod to snap from one to the opposite side of the sphere, jumping from a higher to a lower energy state. In Figure 10, an ideal scanning sweep C (orange: $\psi =-\frac{\pi }{2}$) is outlined and overlaid with the alternative solutions (blue: $\psi =+\frac{\pi }{2}$).

Since the arrangement of deformation states is symmetric, the blue configurations might also be interpreted as an ideal backward sweep in negative direction of *y*. It can be seen that for most clamping positions, except for those exceeding the area below the sphere, there are two configurations (orange and blue) as already observed analyzing Equation (Equation 28). However, from a physical point of view, it is impossible that, e.g., scanning sweep C starts with a blue configuration due to the scanning direction from left to right. This raises the question, at which point of a real-world scanning sweep, the rod might snap from an orange to a blue configuration. Therefore, the elastic deformation energy
$$E=\frac{1}{2}\int_{0}^{s_{1}}\kappa {\left(s\right)}^{2}ds$$
of each deformation state of the rod was calculated. For the scanning sweep in positive *y*-direction, the blue configuration is in a higher energy state than the corresponding orange configuration until the clamping reaches the position $x_{0}=y_{0}=0$. At $x_{0}=y_{0}=0$, the second argument of Equation (Equation 28) is not defined. Practically, there is an infinite number of possible deformation planes $E_{\psi }$ with $\psi_{i}\in \left(-\pi ,\pi \right]$ due to the rotational symmetry of the object. For this particular case, all possible deformation states have the same states of energy and are, therefore, indifferent equilibria. As $y_{0}$ further increases, the blue configurations are in a lower energy state than the orange ones. Therefore, the rod tends to snap to the opposite side of the sphere when $y_{0}$ crosses zero. However, as can be seen in the support reactions of scanning sweep C, the jump occurs long after $y_{0}$ was crossing zero. This is presumably due to the fact that friction has a stabilizing effect in a way that the frictional forces counteract with lateral slip of the rod.

Figure 11 shows the support reactions from Figure 9, which were transformed to the (u,v,w)-coordinate system in the following way:(29)$$\begin{array}{c} \left(\begin{array}{c} f_{0u}\\ f_{0v}\\ f_{0w} \end{array}\right)=T\left(\psi \right)\cdot \left(\begin{array}{c} f_{0x}\\ f_{0y}\\ f_{0z} \end{array}\right),\;\left(\begin{array}{c} m_{0u}\\ m_{0v}\\ m_{0w} \end{array}\right)=T\left(\psi \right)\cdot \left(\begin{array}{c} m_{0x}\\ m_{0y}\\ m_{0z} \end{array}\right) \end{array}$$

In doing so, the angle ψ was calculated using the support reactions $m_{0x}$ and $m_{0y}$ exploiting Equation (Equation 23). The reaction forces $f_{0u}$ (green), $f_{0v}$ (orange) and $f_{0w}$ (blue) during the scanning sweeps A–E are shown on the left side, the reaction moments $m_{0u}$ (green), $m_{0v}$ (orange) and $m_{0w}$ (blue) in the middle and the orientation angle ψ of the deformation plane $E_{\psi }$ on the right side of Figure 11. Theoretically, the components $f_{0w}$, $m_{0u}$ and $m_{0v}$ are zero due to the plane bending assumption. The experimental data of these components show only a slight deviation from the theoretical values.

The overall accordance of the remaining simulation and experimental data ($f_{0u}$, $f_{0v}$ and $m_{0w}$), which is related to the plane bending problem, is even better than in Figure 9. On the one hand, this is apparent from the maximum values of the support reactions. On the other hand, the distortion in the $y_{0}$-direction of the support reactions in Figure 9 is less pronounced in Figure 11. The support reactions are approximately symmetrical with respect to a vertical axis at $y_{0}=0$. The components $f_{0u}$ and $m_{0w}$ are always positive while $f_{0v}$ is always negative. Again, scanning sweep C stands out of Figure 11 due to the jump of the wire from one to the opposite side of the sphere, which was discussed above. In contrast to Figure 9, the support reactions of sweeps A and E, as well as B and D, are approximately the same. This observation indicates that measuring only components $f_{0u}$, $f_{0v}$ and $m_{0w}$ with respect to the deformation plane would not be sufficient to uniquely identify the corresponding scanning sweep. Instead, the angular information on the right side of Figure 11 must be taken into account in order to distinguish sweeps A and E and B and D, based on the support reactions for reliable reconstructions in Section 3.2.

The orientation angle ψ in Figure 11 is determined from the measured and simulated support reactions using Equation (Equation 23). In doing so, both the reaction forces $f_{0x}$ and $f_{0y}$ as well as the reaction moments $m_{0x}$ and $m_{0y}$ are evaluated. The green curves (based on the measured forces) are almost perfectly aligned (obscured) with the orange curves (based on the measured moments). At the beginning and the end, both curves are characterized by strong noise due to the noise of the support reactions before and after contact, which reappear in ψ. Glancing at angle ψ during scanning sweep C, it can be seen that, as the wire jumps from one to the opposite side of the sphere, the angle ψ jumps from $\psi =-\frac{\pi }{2}$ to $\psi =\frac{\pi }{2}$—compare to Figure 10. For the sweeps A, B, D and E, the blue curves, which are based on the reaction forces resulting from simulations, are identical with the black curves, resulting from evaluating the simulated reaction moments. For scanning sweep C, the blue curve deviates from the black curve in a way that it jumps from $\psi =-\frac{\pi }{2}$ to $\psi =\frac{\pi }{2}$ while the black curve remains at $\psi =-\frac{\pi }{2}$. Although this jump seems to resemble the experimental result, it occurs for a different reason: the simulated scanning sweep does not include any jump of the rod—see Figure 7a. Instead, the jump in ψ occurs as a result of a change in sign of α as the contact point crosses the lowest point (south pole) of the sphere. For scanning sweep C, this results in an error when determining ψ based on the reaction forces (blue curve). In contrast, using the reaction moments (black curve), the change in the sign of α does not inhibit a correct determination of ψ. Even though exploiting the forces and moments might lead to different results for ψ, the experimental data (orange and green curves) coincides well. This is due to the fact that the rod snaps around the sphere before the contact point crosses its south pole (before a change in the sign of α). However, a change in the sign of α occurred in experimental scanning sweeps along 2D object contours (no lateral curvature). Therefore, in general, the evaluation of the clamping moment is more reliable to determine the orientation angle ψ. Finally, it can be seen that, for each scanning sweep, the angle based on the measured data lags behind that based on the simulation. This offset presumably results from frictional effects, which are already apparent in the support reactions in Figure 9. The lag in the experimental data of ψ seems to have a corrective effect when transforming the support reactions from the (x,y,z) to the (u,v,w)-coordinate system using Equation (Equation 29)—compare Figure 9 and Figure 11.

### 3.2. Object Shape Reconstruction

Obviously, Figure 9 and Figure 11 do not directly provide any information regarding the scanned object. Therefore, the simulation and experimental quantities are used for object shape reconstruction, as described in Section 2.1. Within the present paper, we use the data in the (x,y,z)-coordinate system from Figure 9, as provided by the used sensor.

**Remark** **5.**
*In general, the data from Figure 11 is equally suitable for object reconstruction and might be chosen as observable but only if the orientation angle ψ defining the (u,v,w)-coordinate system is measured as well. Thus, only four observables, namely $f_{0u}$, $f_{0v}$, $m_{0w}$ and ψ, are sufficient for object reconstruction.*


Initially, we evaluated only those 27 scanning sweeps with smallest object distance h=25 mm. The experiments showed that, for this object distance, the size of the scanned part of the object was the largest and the support reactions had the best signal to noise ratio. In this way, we aimed to investigate, whether a single object distance is sufficient for object reconstruction. In the first row of Figure 12, the reconstructed sequences of contact points based on 27 simulated scanning sweeps with an object distance of h=25 mm are outlined, where Figure 12a is an isometric and Figure 12b is a top view.

Each reconstructed sequence of contact points resulting from a single simulated sweep is connected with a line. The colors of the points are arbitrarily chosen and are solely aimed for an easy distinction between adjacent contact point sequences. As the theoretical results indicate, object scanning using a single object distance is sufficient to provide a variety of contact points, approximating the shape of the sphere. However, there is a large gap in the reconstructed area. It is crossed by the reconstructed sequence of contact points resulting from the special scanning sweep C ($x_{0}=0$). The theoretical reconstruction errors of all points are negligible (within the numerical boundaries). The symmetry of the support reactions (see Figure 9) reappears in the reconstructed sequences of contact points, which are axially symmetric with respect to the *x*-axis (except that resulting from scanning sweep C ending with a snap-off). In addition, those contact sequences resulting from scanning sweeps with the same distance $|x_{0}|$ from the coordinate origin are symmetrical in pairs, with respect to the *y*-axis. Having in mind that the step-size $\Delta y_{0}$ for the displacement of the clamping position $P_{0}$ is constant for all simulations, it is important to note that the reconstructed points are not equidistantly distributed. Instead, the distances between the points increase in the middle of each sequence—see, e.g., Figure 12a (blue). This effect is reinforced the closer the scanning trail is to the coordinate origin, i.e., the smaller $|x_{0}|$. In particular, this is evident in the innermost sequences, which narrows the reconstruction gap. Even though the presented mechanical model is a quasi-static one, this fact may be a hint at the contact point velocity during scanning. It can be expected that the ratio between the displacement of the contact point and the displacement of the clamping position might be the same as that between the contact point velocity and the clamping velocity. Thus, greater distances between the reconstructed points might be interpreted as higher contact point velocities. This is consistent with our observations during the experiments: the contact point starts moving slowly and gets increasingly fast in the middle of the scanning sweep until it finally slows down again.

Figure 12c,d show the reconstructed points based on the measured data of 27 scanning sweeps (h=25 mm) superimposed with the original object surface. The color of each point denotes the reconstruction error in mm, which is defined as the closest distance between the point and the object surface. The error is directly evident from the color bar in Figure 12d. It can be seen that the maximum reconstruction error after expunging the outliers is 1.58 mm or approximately 3.2% of the wire length. A positive sign of the reconstruction error indicates that a contact point is outside and a negative sign that it is inside of the original object. Having the scanning direction in mind (positive *y*-direction), it is obvious that positive errors occur, especially at the beginning and the end of the scanning sweep, when the deformation of the wire is small. These errors are not surprising, since small wire deformation results in a bad signal to noise ratio. Negative errors increasingly occur as the wire bends in the *x*-direction. In Figure 12d the outer reconstructed sequences seem fairly axial symmetric with respect to the *x*-axis. In addition, the pairwise arrangement of sweeps with the same lateral displacement $|x_{0}|$ are symmetrical to some extent. However, it is striking that the reconstruction error seems asymmetric in the sense that negative errors are greater for x>0. Figure 9 has already shown that the support reactions resulting from scanning sweeps with $x_{0}>0$ (sweep D and E) were more affected by stick-slip effects, which probably cause the asymmetry of the reconstruction error. The jump of the rod from one side to the opposite side of the sphere during scanning sweep C is apparent, looking at the innermost sequence of reconstructed contact points, which start at x=0. At the beginning of the scanning sweep, all contact points approximately lie in the x-z-plane, which indicates that only longitudinal slip occurred. Subsequently, the contact point begins to move slightly laterally in a *x*-direction. Finally, the sequence of contact points breaks off due to the jump of the rod to the opposite side of the sphere. It is interesting that this behavior is not only apparent for the innermost scanning sweep but also for some sweeps with a small lateral displacement $|x_{0}|$. This could be related to the large distances between the reconstructed points when simulating scanning sweeps with a small lateral displacement $|x_{0}|$.

Perhaps surprisingly, the distortion in the $y_{0}$-direction between the simulated and measured data in Figure 9, which is presumably friction-induced, does not appear to affect the outcome of the object reconstruction. As the color bar shows, there is no significant distortion between the reconstructed points and the original object. In order to understand this result, the effect of friction during object scanning is analyzed in more detail. One key assumption of the mechanical model is that the scanning sweep takes place in the absence of friction. Therefore, in theory, the contact force is always orientated in direction of the outward-pointing normal vector of the object surface—see Figure 13.

However, in a real-world scanning sweep, friction causes an additional tangential force component opposed to the direction of contact point movement. Therefore, the resulting contact force vector no longer coincides with the surface normal. For a given clamping position $P_{0}$, this leads to a different deformation plane $E_{\psi }^{\prime }$ (and consequently to another contact point $P_{1}^{\prime }$), than that, which would have been achieved in the absence of friction. This explains the deviations between the simulated and experimental data in Figure 9 as well as the distortions of the orientation angle ψ in Figure 11. If the measured support reactions are used for contact point localization, the resulting contact force ${\vec{f}}^{\prime }$ can be determined, but not decomposed into its normal and tangential components ${\vec{f}}_{n}$ and ${\vec{f}}_{t}$. Therefore, the normal vector ${\vec{n}}_{1}^{\prime }$ of the surface remains unknown. However, as the single force assumption is not violated, the rod still bends in a plane. Using Equation (Equation 23) exploiting the measured, friction effected support reactions, the orientation $\psi^{\prime }$ of the friction-affected deformation plane can be determined. Thus, even in the presence of friction, the object reconstruction problem reduces to the plane once, discussed in Section 2.1. It is important to understand that, even though a point $P_{1}^{\prime }$ which is reconstructed in the presence of friction (experiment) is different from the point $P_{1}$, which would have been reconstructed in the absence of friction (simulation), both points $P_{1}^{\prime }$ and $P_{1}$ lie on the original object surface.

This observation indicates that the object reconstruction is less sensitive to frictional effects than perhaps expected, which highlights an important characteristic of the measuring principle. However, stick-slip effects might cause reconstruction errors due to the fact that they induce dynamical effects.

In Figure 12e,f the reconstruction algorithm presented in [27] was applied to the data in Figure 12c,d. Using implicit B-splines, the algorithm fits a surface to a 3D cloud of points accompanied by normal vectors. In general, the involvement of the surface normals improves the reconstruction result as they contain additional information about the local surface orientation. However, as discussed above, the reconstructed force vector no longer coincides with the normal vector of the surface in the presence of friction. Therefore, considering the reconstructed contact force vector as a normal vector is only a rough approximation and only appropriate in the case of little friction. This raises the question of how friction might be estimated during object scanning, which is discussed in Section 4. In Figure 12e,f the point clouds are overlaid with the reconstructed tactile image of the scanned object (gridded surface). It is obvious that the algorithm interpolates between and extrapolates beyond the reconstructed points. In our experiment, 27 scanning sweeps with a single object distance are sufficient to reconstruct a significant part of the object. Visually comparing the tactile image from Figure 12e with the original object surface in Figure 12c, it can be seen that, despite frictional effects during the experiment, the evaluation of the force vectors for generating the tactile image does not result in significant reconstruction errors.

Even though a large part of the object is reconstructed in Figure 12f there remains a reconstruction gap. Evaluating all 135 scanning sweeps with different object distances, this reconstruction gap can be filled to a large extent as shown in Figure 14. Even though there still remains an area with a lack of information due to the limited reachability of the rod, the extent of the reconstruction is probably sufficient for many cases of application. Otherwise, the reachability of the rod might be improved by either changing the scanning direction or changing from a sweeping to a tapping scanning strategy. Returning to the biological paragon, this reflects the behavior that can be observed in rats during environmental exploration [29].

## 4. Conclusions

In this paper, we presented a vibrissa-inspired sensor principle for 3D object scanning and reconstruction. An artificial vibrissa was modeled as a one-sided clamped Euler–Bernoulli bending rod with a constant circular cross-section. The clamping reactions (observables) at the base of the rod were the only quantities used to draw conclusions about the object shape, as in biology. For object scanning, the clamping position of the rod was translationally shifted relative to a smooth and convex 3D object surface, whereby the authors are aware of this limitation due to the fundamental analytical description. The scanning procedure was considered quasi-statically. As the rod slid over the object’s surface, undergoing large bending deflection, both longitudinal and lateral slip occurred. It was shown that, due to the material and geometric isotropy of the rod, a single contact force during object scanning causes plane bending of the rod in some unknown deformation plane. The focus of the investigation was on both generating the clamping reactions during object scanning theoretically and using these quantities to reconstruct the contact position in space with a focus on an experimental validation. During scanning, a distinction was made between tip and tangential contact phases. For simulating scanning sweeps and generating the clamping reactions, each contact phase resulted in an individual boundary-value problem. Repeatedly solving these boundary-value problems for different clamping positions, the unknown contact parameters (contact force, its direction and position along the rod) and finally, the support reactions were generated theoretically. The mechanical model and the simulation algorithm was validated using an experimental setup consisting of a 50 mm spring steel wire attached to a 3D force-torque transducer in order to measure the full set of six support reactions at the base of the wire. A glass sphere with a diameter of 60 mm was used as a test object to perform 135 scanning sweeps varying the scanning trails and the object distance.

The measured support reactions showed a good overall accordance with the simulated data, but the measured data was significantly distorted due to frictional effects. One scanning sweep, which was performed directly below the center of the sphere, turned out as a plane special case without any lateral slip. For this sweep, the simulation outcome differed from the observations during the experiment, which, however, could be explained by taking the deformation energy into account. Finally, both the generated and measured clamping reactions were used to reconstruct a sequence of contact points with corresponding normal vectors. The fact that these data include the entire scope that can be sensed, e.g., by using a laser range finder emphasizes that the presented tactile sensor principle is highly suitable to be used in a complementary way with optical sensors.

A large part of the sphere was reconstructed based on solely 27 scanning sweeps with a single object distance. While the simulated reconstruction error was within the numerical boundaries, the maximum experimental error was 1.58 mm or 3.2% of the wire length. This error relates to each individual reconstructed contact point and is not expected to change significantly for different object sizes or shapes. Despite the fact that we did not actively prevent any lateral slip and that the measured signals at the base of the wire are several orders of magnitude smaller than in [23], the resolution of our test setup is in the same order of magnitude as in [23]. Even though frictional effects caused deviations between the simulated and measured clamping reactions, they had no significant impact on the reconstructed contact points. In this way, the sensor principle turned out to be the friction invariant, to a large extent, with respect to the reconstructed points, but not to the reconstructed surface normals. An existing approach from 3D computer vision [27] was used for fitting a surface to the set of reconstructed points accompanied with normal vectors. The latter are determined by evaluating the contact force vectors which, however, only correspond to the normal vectors of the surface in the absence of friction. Therefore, including the contact force directions for fitting a surface to the cloud of reconstructed points only provides satisfying results for scanning sweeps with low friction.

Within the present paper the entire set of mechanical signals at the base of a rod was considered deliberately for a detailed mechanical investigation of the connection between these signals and the according contact points in space. As a result, it could be shown, that the number of required observables for object reconstruction can be reduced from six (three forces and three moments) to four (two forces and two moments, or even two forces, one moment and one angular information). From an analytical point of view, our model suggests that these four quantities are the minimum number of observables required for 3D reconstruction. However, due to the complex nature and the large dimensions of the force-torque transducer used within the present paper, the sensor principle is not yet suitable for array implementation, i.e., using a larger number of vibrissa-based sensors on a robot, collecting and exploiting overlapping information during object scanning.

Future studies should focus on estimating frictional effects during object scanning with the final goal to decompose the detected contact forces in a surface normal and tangential component, respectively, and, thus, to improve the reconstruction result. One way to address this problem might be exploiting the twist moment component at the clamping, as it seems to be directly related to the tangential force component. Another approach to estimate friction might be achieved in the following way. As discussed in the present paper, the components of the contact force vector cannot be decomposed in the given way in the presence of friction. However, the reconstructed contact points might be used to determine the local surface orientation, which might allow us to estimate the surface normal direction and thus to decompose the components of the contact force.

Furthermore, the presented model should be extended for scanning 3D surfaces including planes, edges and vertices in future work. This objective has already been partially addressed, see [25]. Subsequently, the model might serve as an important basis for future parameter studies with the goal to optimize the sensor design without the necessity of performing a wide range of experiments. One optimization aspect of particular importance is the used force-torque transducer. Although, these sensors nowadays have increasingly higher resolution and are becoming lower in costs, the latter are still far too high for a widespread application in terms of tactile sensing. Therefore, the redesign and re-scaling of the used sensor must be a key topic to address in future work. Returning to the biological paragon, the FSC of animals might give further hints for an appropriate constructive implementation of the used sensors at the base of a rod. According to [30], mechanoreceptors show a longitudinal and radial distribution inside of the FSC. There, it was observed that all mechanoreceptors exhibit a strong “angular tuning”, which means that their response is highly dependent on the angle of deflection, which, in the context of the present work, is the orientation angle ψ. Therefore, a radial arrangement of sensors at the base of a rod should be investigated in future work. 

## Figures and Tables

**Figure 1 sensors-21-01572-f001:**
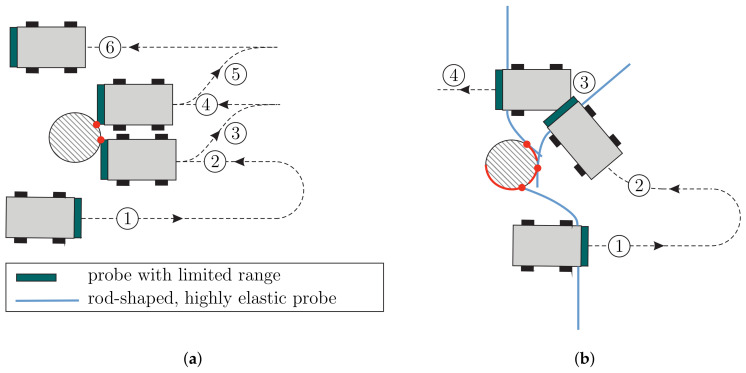
Mobile robots equipped with tactile sensors exploring using different probes to explore their environments: (**a**) robot with a dual switch or a skin-like sensor with limited range—(1) robot passing the object without sensor contact, (2) first collision with the object leads to the detection of a single discrete point, (3) moving backwards/ repositioning, (4) second collision with the object leads to the detection of a second discrete point, (5) see (3), (6) obstacle avoidance completed; (**b**) robot with a rod-shaped, highly flexible sensor—(1) passing the object witch continuous sensor contact (detecting a sequence of contact points), (2) using information from (1) for making the next contact, (3) wall-following from (2) to (3) by keeping the signals constant, (4) obstacle avoidance completed.

**Figure 2 sensors-21-01572-f002:**
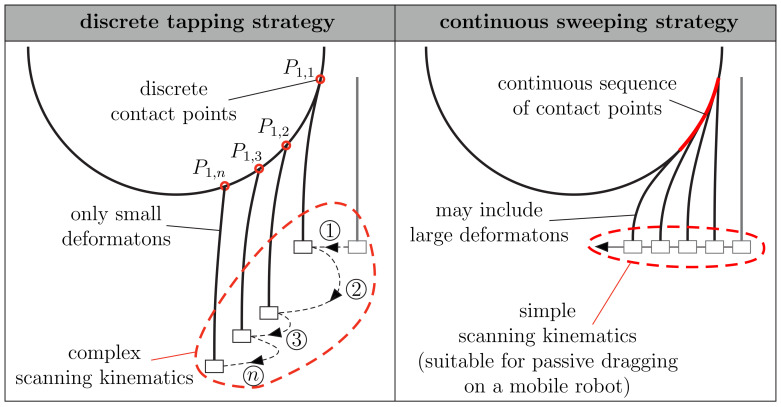
Comparison of two different object scanning strategies.

**Figure 3 sensors-21-01572-f003:**
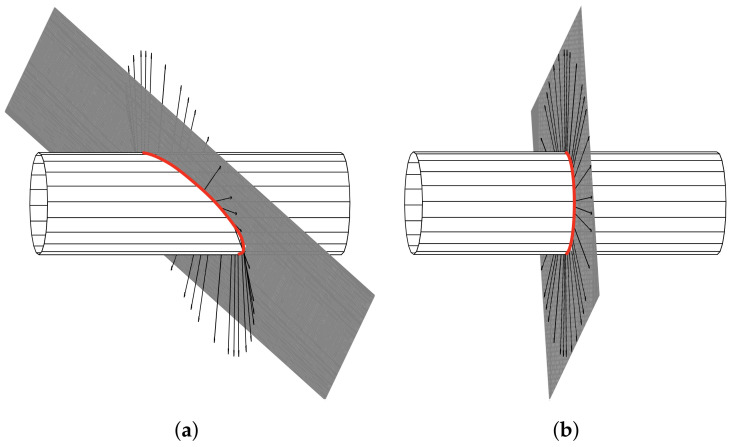
Intersection curve (red) between an exemplarily scanned object (cylinder) and the plane of rotation/translation (scanning plane-gray) of the rod: (**a**) arbitrary arrangement; (**b**) special arrangement resulting in a plane scanning sweep.

**Figure 4 sensors-21-01572-f004:**
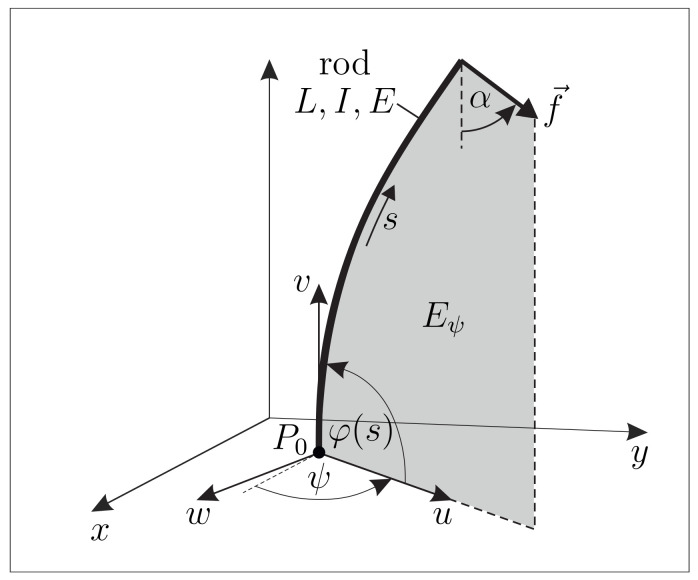
Mechanical model of the isolated artificial vibrissa in the inference of some contact force.

**Figure 5 sensors-21-01572-f005:**
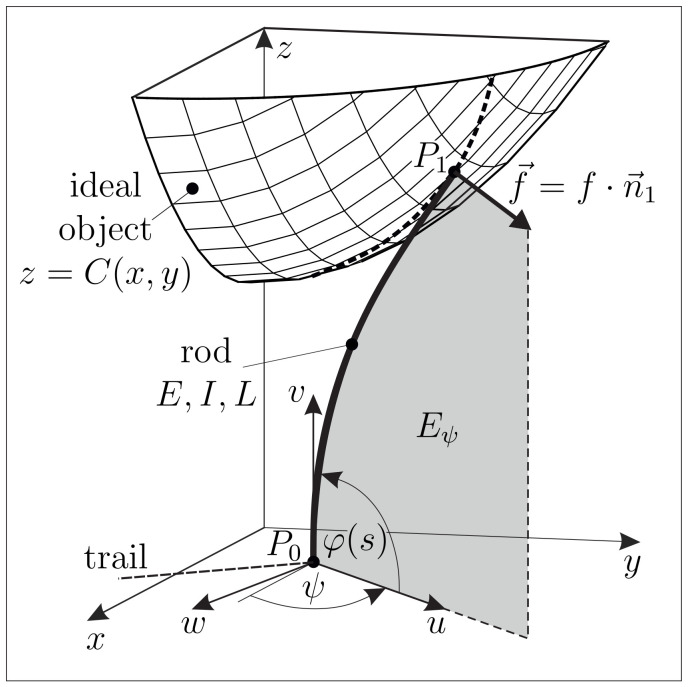
Mechanical model of an artificial vibrissa in ideal contact with an object.

**Figure 6 sensors-21-01572-f006:**
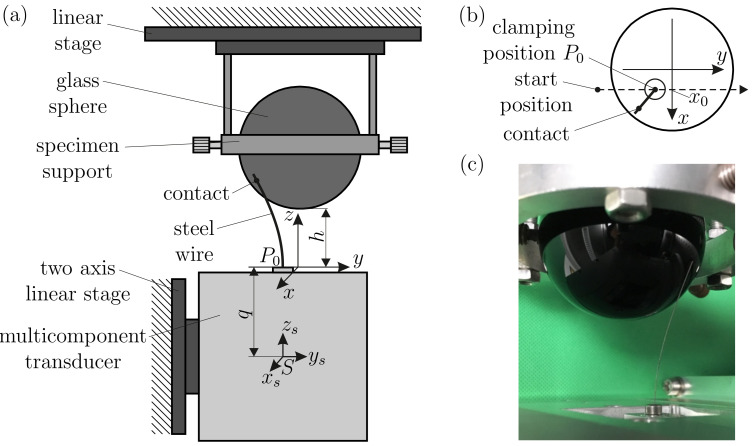
Experimental setup: (**a**) schematic representation of the experiment; (**b**) top view on (**a**); (**c**) one-sided clamped wire in tangential contact with a glass sphere.

**Figure 7 sensors-21-01572-f007:**
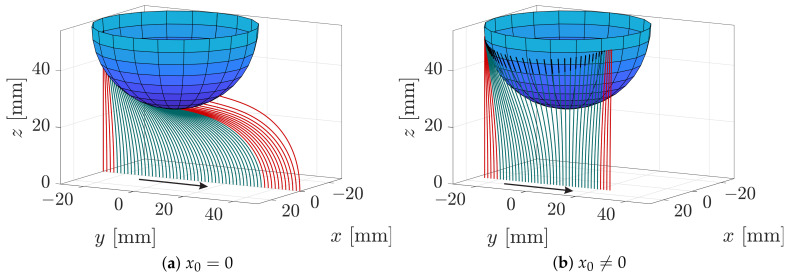
Simulated sequence of elastic lines during two scanning sweeps along a 60 mm sphere (red—tip contacts, green—tangential contacts, black—undeformed end of the wire.

**Figure 8 sensors-21-01572-f008:**
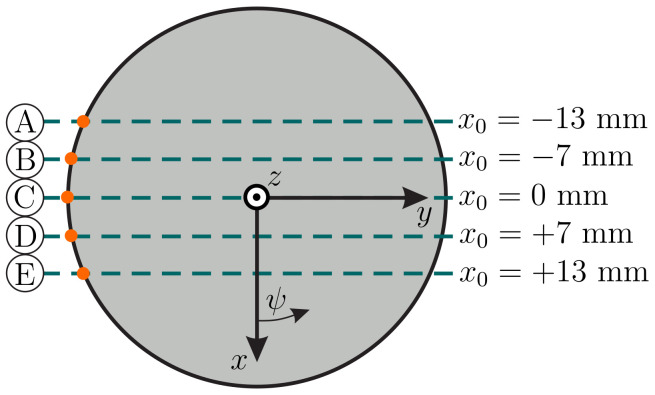
Schematic representation of five exemplary scanning trails (**A**–**E**).

**Figure 9 sensors-21-01572-f009:**
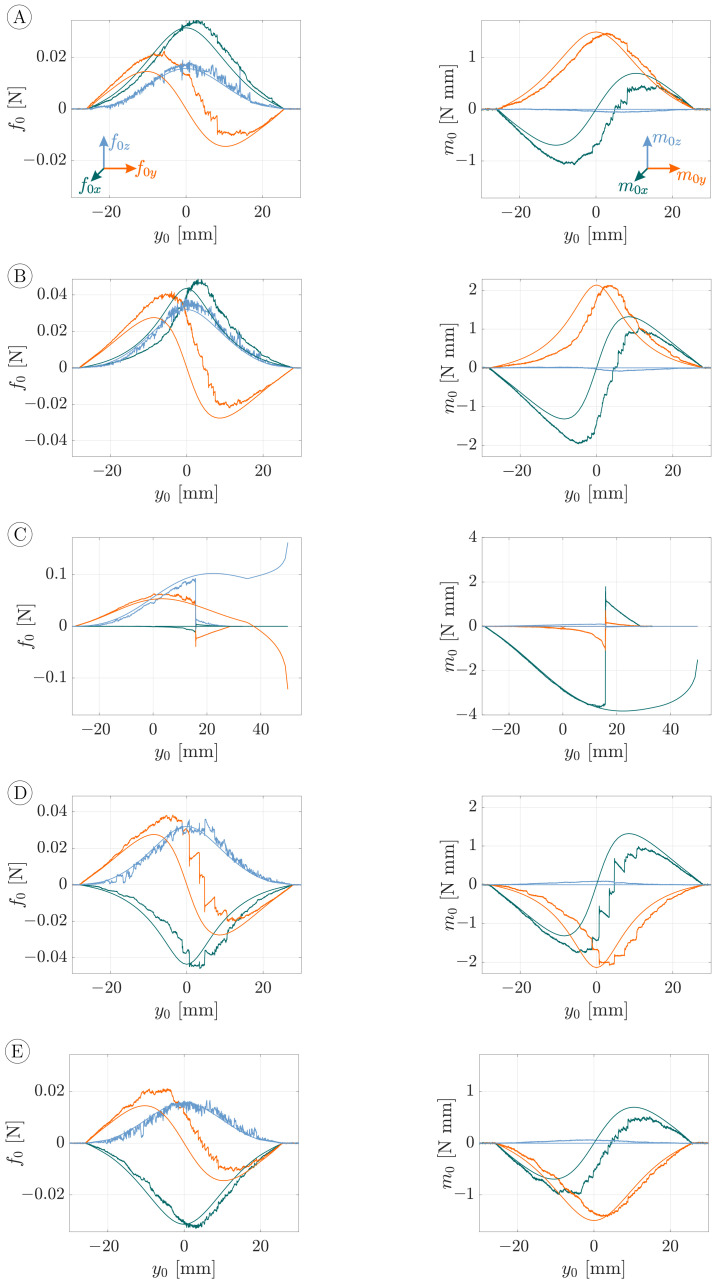
Simulated and measured observables, represented by smooth and noisy data, respectively, during object scanning of a 60 mm glass sphere. Each row results from object scanning according to the scanning sweeps (**A**–**E**), schematically shown in Figure 7, and includes the following diagrams: reaction forces on the left and reaction moments on the right.

**Figure 10 sensors-21-01572-f010:**
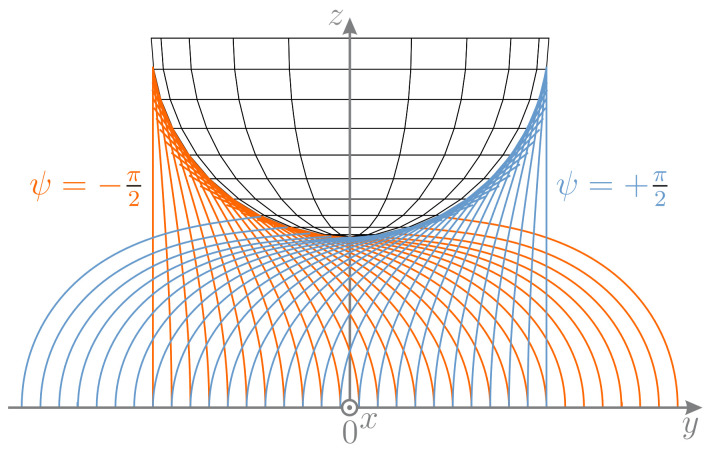
Entirety of solutions for $x_{0}=0$ with some of the blue ($\psi =+\frac{\pi }{2}$) and orange ($\psi =-\frac{\pi }{2}$) solutions sharing the same clamping position $P_{0}$.

**Figure 11 sensors-21-01572-f011:**
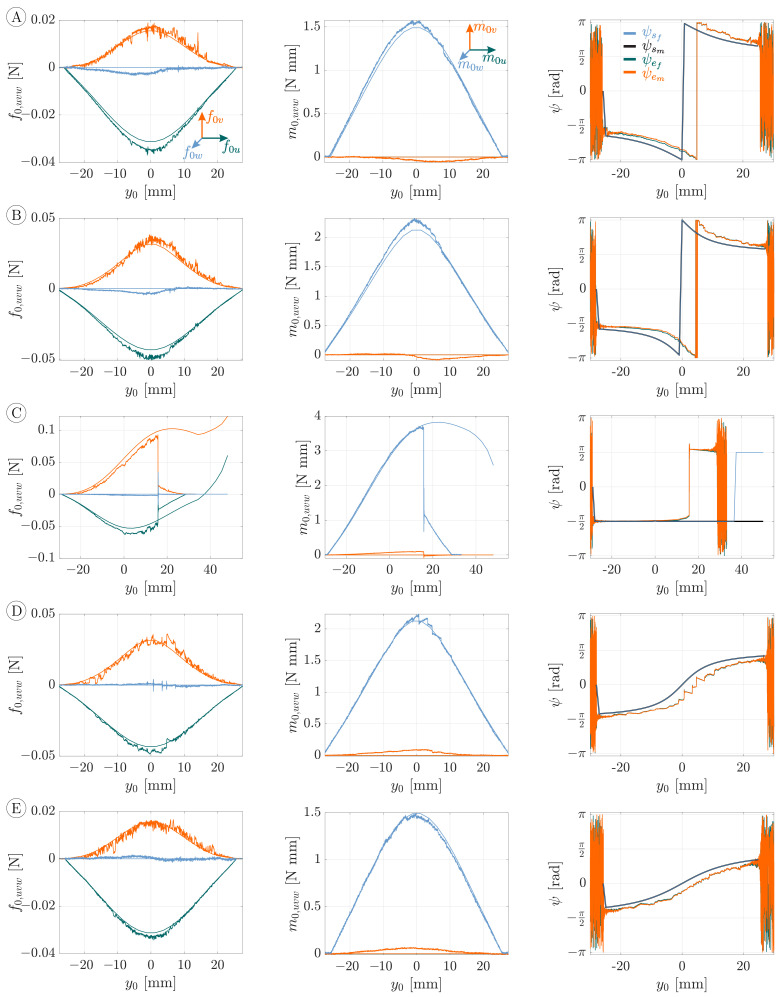
Simulated and measured observables, represented by smooth and noisy data, respectively, during object scanning of a 60 mm glass sphere. Each row results from object scanning according to the scanning sweeps (**A**–**E**) schematically shown in Figure 7 and includes the following diagrams: reaction forces on the left, reaction moments in the middle and orientation angle ψ of the deformation plane calculated using Equation (Equation 23) on the right ($\psi_{s_{f}}$ based on the simulated reaction forces in blue, $\psi_{s_{m}}$ based on the simulated reaction moments in black, $\psi_{e_{f}}$ based on the measured reaction forces in green and $\psi_{e_{m}}$ based on the measured reaction moments in orange). Note that, for the $y_{0}$-ψ-diagram, both the green and orange curves, as well as the black and blue curves, almost coincide and, therefore, obscure each other to a large extent.

**Figure 12 sensors-21-01572-f012:**
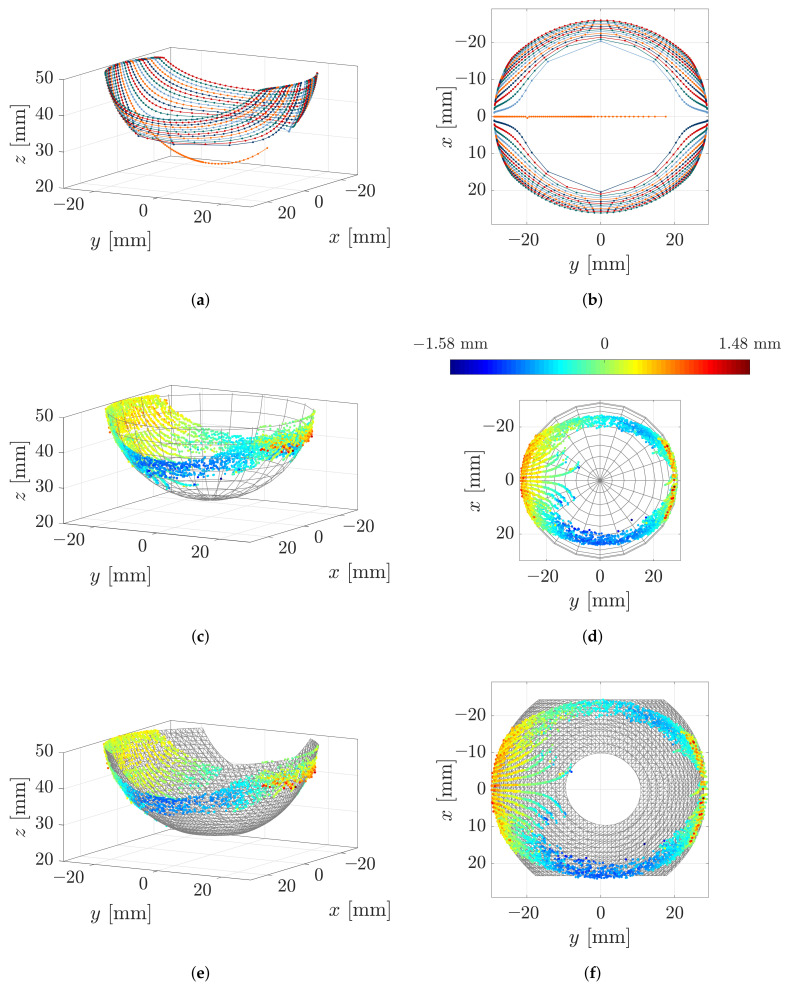
Object reconstruction: (**a**,**b**) reconstructed contact points based on simulated data (each connected sequence of points results from one scanning sweep); (**c**,**d**) reconstructed contact points based on experimental data superimposed with the original object shape (the color of each point indicates the reconstruction error); (**e**,**f**) fitted surface providing a tactile image based on the reconstructed contact points and the corresponding force directions.

**Figure 13 sensors-21-01572-f013:**
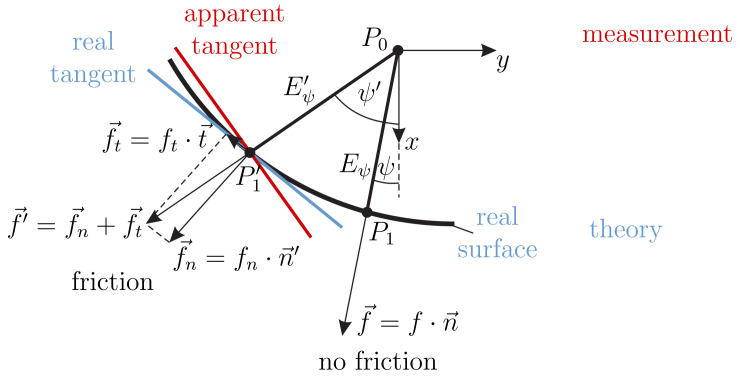
Schematic representation of two deformation states at the same clamping position $P_{0}$ where one is affected by friction and the other one is the reference state without friction. In the presence of friction, the quantities are marked with an apostrophe—${\vec{f}}_{t}$ is the tangential force component with the magnitude $f_{t}$ and direction $\vec{t}$, ${\vec{f}}_{n}$ is the normal force component with the magnitude $f_{n}$ and direction ${\vec{n}}^{\prime }$.

**Figure 14 sensors-21-01572-f014:**
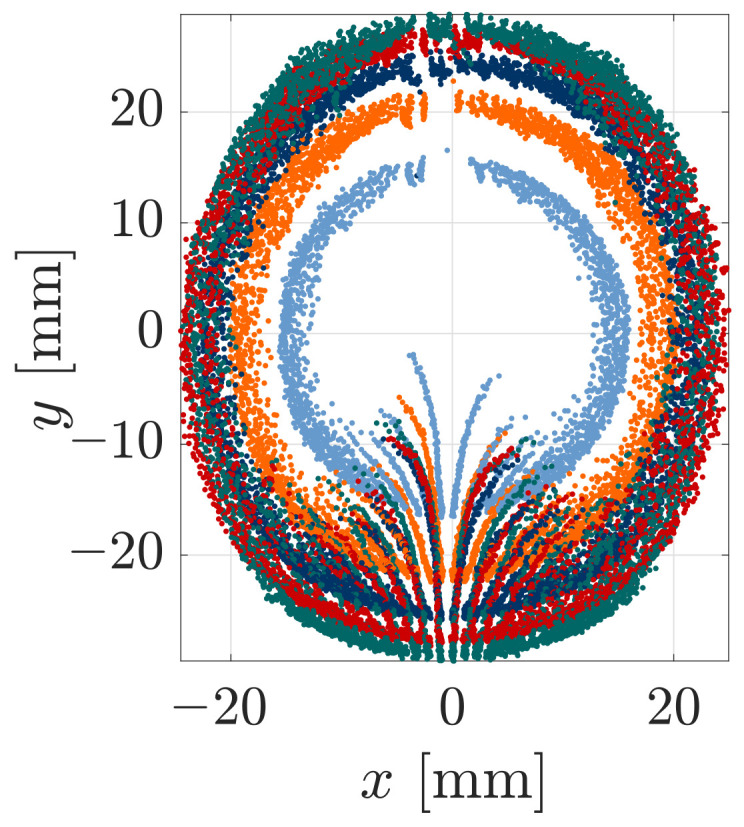
Reconstructed contact points based on the data from all 135 scanning sweeps (object distance: 25 mm in green, 30 mm in red, 35 mm in dark blue, 40 mm in orange and 45 mm in blue).

## Data Availability

All data can be requested by contacting the corresponding author.

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
