# Peer review of "A Vibrissa-Inspired Highly Flexible Tactile Sensor: Scanning 3D Object Surfaces Providing Tactile Images"

_sensors, 2021, doi:10.3390/s21051572_

Round 1
Reviewer 1 Report
This paper presented a vibrissa-inspired sensor concept for 3D object scanning and reconstruction that can be used in mobile robots. They use flexible rods to scan 3D objects, collect data points through force and torque sensors, and finally reconstruct 3D objects. For this manuscript, a few questions should be addressed explicitly.
- The flexible rod is used to avoid obstacles. How to install the flexible rod on the robot car to ensure that the robot does not collide with obstacle and does not have too much margin (means that the robot can pass through, but the flexible rod detects the obstacle and mistakenly thinks that it cannot pass).
- This manuscript focuses on scanning the sphere. When scanning objects of different shapes, is the detection accuracy different? What type of target objects have higher detection accuracy?
- For the multicomponent force and torque transducer, what are their respective ranges, how about the detection accuracy. Does flexible rods with different elastic modulus E affect the detection accuracy?
- For the situation of both longitudinal and lateral slip occurred, does the uneven relative sliding velocity due to the friction on the surface of the scanned object affect the model reconstruction?
- The scanning in this manuscript is to scan the target object every certain distance, which means that if the robot wants to scan the shape of the obstacle when passing through the obstacle, it needs to equip several flexible rods. How does this differ from the experimental conditions in this paper?
- The data points used to reconstruct the 3D model are not equally spaced because the scanning speed is first slow and then fast and then slow. Does this have any effect on 3D reconstruction? How about for constant speed moving?
Reviewer 2 Report
The authors presented a detailed study of a whisker-like scanning approach for 3D object surface reconstruction.
The authors reported their work in great details, which provide a full picture of the work. The reported approach were described well, experiments are well-designed, and the results fairly discussed and analyzed.
However, considering it's a journal article, not a technical report, the logic and structure of the manuscript should be improved to make it easy to follow and catch the key contributions. The manuscript is probably too long, containing too much non-essential context. For example, some detailed descriptions and equation derivation can be move to an appendix section or as supplementary information.
My major comments are:
1. From a scientific study point of view, I do believe the work presented some insight and have its unique values. But I'm skeptical of the value of the proposed approach in real applications. In the conclusion, the authors commented that that multi-DoF force/Torque sensors becomes much cheaper for such purpose. But I guess it's still way too expensive to use a 6-axis F/T sensor (3-10k euro level for these I knew) to monitor the reaction force/torque of the whisker during the scanning. The majority of the robotic arms are not even equipped with 6 axis F/T sensors because of the high-cost. Maybe an algorithm/approach that has significant lower requirements of hardware would be more promising. Have the authors considered these emerging soft hair/whisker sensing arrays? How the scanning process and the proposed approach can be applied there? These sensors might not have a 6-DoF signals output (only pressure, or tri-axis force/deformation), but probably more efficient and cost-effective in real applications.
2. In Figure 6, maybe gradient color/grey filling can be used to make the illustration better. It’s a bit difficult to tell which is concave or convex from the presented image?
3. In figure 11&13, it would be much better to direct add the label/legend of the curves on the figures, instead of describing them in figures' caption.
4. Information of the force/torque measurement system/sensor should be provided. (Manufacturer, model number, etc.)
Reviewer 3 Report
Dear authors,
This reviewer wants to congratulate you on the work done. The work presents a vibrissae-based object reconstruction method that shows the potential of this option.
This reviewer finds the work well focused and structured and with two relevant conclusions:
- The reconstruction is less sensitive to frictional effects than expected.
- Stick slip effects might cause noticeable reconstruction errors.
To my understanding, only a few very small minor issues require modification:
1- The word "Abstract" has disappeared on the first page.
2- Throughout the text in multiple places the authors talk about future works. Group these comments into a section. It will be more interesting for the readers and will structure the work better.
Kind Regards
Round 2
Reviewer 2 Report
The authors addressed my comments/concerns fairly well.